# VectorMapNet: End-to-end Vectorized HD Map Learning

## Abstract

Autonomous driving systems require a good understanding of surrounding environments, including moving obstacles and static High-Definition (HD) semantic map elements. Existing methods approach the semantic map p·roblem by offline manual annotation, which suffers from serious scalability issues. Recent learning-based methods produce dense rasterized segmentation predictions to construct maps. However, these predictions do not include instance information of individual map elements and require heuristic post-processing to obtain vectorized maps. To tackle these challenges, we introduce an end-to-end vectorized HD map learning pipeline, termed VectorMapNet. VectorMapNet takes onboard sensor observations and predicts a sparse set of polylines in the bird's-eye view. This pipeline can explicitly model the spatial relation between map elements and generate vectorized maps that are friendly to downstream autonomous driving tasks. Extensive experiments show that VectorMapNet achieve strong map learning performance on both nuScenes and Argoverse2 dataset, surpassing previous state-of-the-art methods by 14.2 mAP and 14.6mAP. Qualitatively, we also show that VectorMapNet is capable of generating comprehensive maps and capturing more fine-grained details of road geometry. To the best of our knowledge, VectorMapNet is the first work designed towards end-to-end vectorized map learning from onboard observations.

## 1 Introduction

Autonomous driving system requires an understanding of map elements on the road, including lanes, pedestrian crossing, and traffic signs, to navigate the world. Such map elements are typically provided by pre-annotated High-Definition (HD) semantic maps in existing pipelines (Rong et al., 2020). These methods suffer from serious scalability issues as human efforts are heavily involved in annotating HD maps. Recent works (Li et al., 2021; Philion & Fidler, 2020; Roddick & Cipolla, 2020) explore the problem of online HD semantic map learning, where the goal is to use onboard sensors (*e.g.* LiDARs and cameras) to estimate map elements on-the-fly.

Most recent methods (Roddick & Cipolla, 2020; Yang et al., 2018; Philion & Fidler, 2020; Zhou & Krähenbühl, 2022) consider HD semantic map learning as a semantic segmentation problem in bird's-eye view (BEV), which rasterizes map elements into pixels and assigns each pixel with a class label. This formulation makes it straightforward to leverage fully convolutional networks. However, rasterized maps are not an ideal map representation for autonomous driving, for three reasons. First, rasterized maps lack instance information which is necessary to distinguish map elements with the same class label but different semantics, *e.g.* left boundary and right boundary. Second, it is hard to enforce spatial consistency within the predicted rasterized maps, *e.g.* nearby pixels might have contradicted semantics or geometries. Third, 2D rasterized maps are incompatible with most autonomous driving systems which consume instance-level 2D/3D vectorized maps for motion forecasting and planning.

To alleviate these issues and produce vectorized outputs, HDMapNet (Li et al., 2021) generates semantic, instance, and directional maps and vectorizes these three maps with a hand-designed post-processing algorithm. However, HDMapNet still relies on the rasterized map predictions, and its heuristic post-processing step complicates the pipeline and restricts the model's scalability and performance.

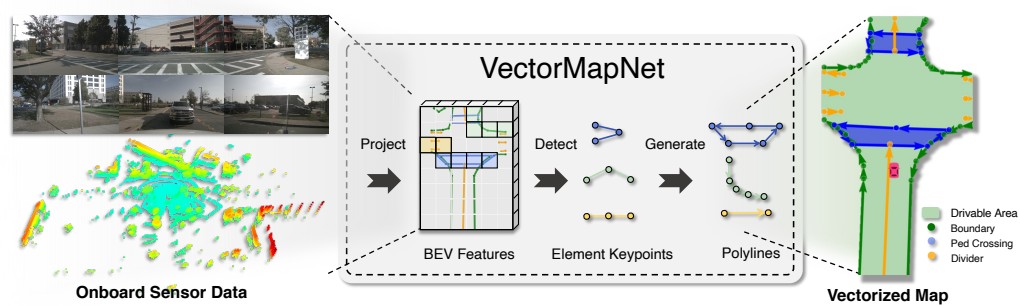

Figure 1: An overview of VectorMapNet. Sensor data is encoded to BEV features in the same coordinate as map elements. VectorMapNet detects the locations of map elements from BEV features by leveraging element queries. The vectorized HD map is built upon a sparse set of polylines that are generated from the detection results. Since polylines have encoded direction information, we can infer semantic information (*e.g.* drivable area) from the polylines. It worth noting that the drivable area is inferred from several disjoint boundaries and is non-trivial to model as one object.

In this paper, we propose an end-to-end vectorized HD map learning model named VectorMapNet, which does not involve a dense set of semantic pixels. Instead, it represents map elements as a set of polylines that are closely related to downstream tasks, *e.g.* motion forecasting (Gao et al., 2020). Therefore, the map learning problem boils down to predicting a sparse set of polylines from sensor observations in our paper. Specifically, we pose it as a detection problem and leverage set detection and sequence generation methods. First, VectorMapNet aggregates features generated from different modalities (*e.g.* camera images and LiDAR) into a common BEV feature space. Then, it detects map elements' locations based on learnable element queries and BEV features. Finally, we decode element queries to polylines for every map elements. An overview of VectorMapNet is shown in Figure 1.

Our experiments show that VectorMapNet achieves state-of-the-art performance on the public nuScenes dataset (Caesar et al., 2020) and Argoverse2 (Wilson et al., 2021), outperforming HDMap-Net and another baseline by at least 14.2 mAP. Qualitatively, we find that VectorMapNet builds a more comprehensive map compared to previous works and is capable of capturing fine details, *e.g.* jagged boundaries. Furthermore, we feed our predicted vectorized HD map into a downstream motion forecasting module, and show the compatibility and effectiveness of the predicted map.

To summarize, the contributions of the paper are as follows:

- VectorMapNet is an end-to-end HD semantic map learning method. Unlike previous works, we pose map learning as an set prediction problem and directly predict vectorized outputs from sensor observations without requiring map rasterization or post-processing.
- Jointly modeling the geometry and topological relations of map elements is challenging. We leverage polylines as primitives to model complex map shapes and decompose the model into two parts to mitigate this difficulty: a map element detector and a polyline generator.
- VectorMapNet achieves state-of-the-art HD semantic map learning performance on both nuScenes and Argoverse2 datasets. Qualitative results and downstream evaluations also validate our design choices.

## 2 VECTORMAPNET

**Problem formulation.** Similar to HDMapNet (Li et al., 2021), our task is to model map elements in a vectorized form using data from onboard sensors, *e.g.* RGB cameras and/or LiDARs. These map elements include but are not limited to : Road boundaries, boundaries of roads that split roads and sidewalks. Typically, they are curves with irregular shapes and arbitrary lengths; Lane dividers, boundaries of the lanes in the road. Usually they are straight lines; Pedestrian crossings, regions with white markings where pedestrians can legally cross the road. Usually they are quadrilaterals. These elements are critical for autonomous driving, but these elements typically have diverse geometries

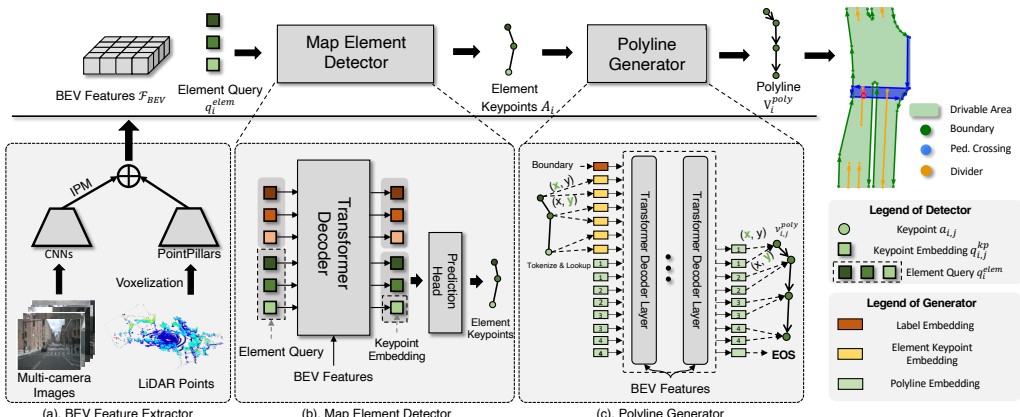

Figure 2: The network architecture of VectorMapNet. The top row is the pipeline of VectorMapNet generating polylines from raw sensor inputs. The bottom row illustrates detailed structures and inference procedures of three primary components of VectorMapNet: BEV feature extractor, map element detector, and polyline generator. Numbers in polyline embeddings indicate predicted vertex indexes.

and semantic meaning. For example, in HD semantic maps, lanes are usually represented as curves, pedestrian crossings are often represented as polygons.

The heterogeneous nature of map elements calls for a unified vectorized representation. We opt to use $N$ polylines $\mathcal{V}^{\mathrm{poly}} = \{V_1^{\mathrm{poly}}, \dots, V_N^{\mathrm{poly}}\}$ as primitives to represent these map elements in a map $\mathcal{M}$. Each polyline $V_i^{\mathrm{poly}} = \{v_{i,n} \in \mathbb{R}^2 | n = 1, \dots, N_v\}$ is a collection of $N_v$ ordered vertices $v_{i,n}$. In practice, we converts vector HD maps from different public datasets to polylines by applying the Ramer–Douglas–Peucker algorithm (Ramer, 1972).

**Why Polyline?** Using polylines to represent map elements has three main advantages: (1) HD maps are typically composed of a mixture of different geometries, such as points, lines, curves, and polygons. Polylines are a flexible primitive that can represent these geometric elements effectively. (2) The order of polyline vertices is a natural way to encode the direction of map elements, which is vital to driving. (3) The polyline representation has been widely used by downstream autonomous driving modules, such as motion forecasting (Gao et al., 2020).

**Method overview.** We formulate this task as a sparse set detection problem. Specifically, we represent a map $\mathcal{M}$ by a sparse set of polylines, and the task is to learn a model that extracts information from sensors to predict these primitives for representing the semantic map.

First, we map sensor data from sensor-view to a canonical BEV representation $\mathcal{F}_{\mathrm{BEV}}$. Then the remaining task is to model polylines based on $\mathcal{F}_{\mathrm{BEV}}$. However, map elements exhibit complicated and diverse structural and location patterns, learning both of them jointly can be challenging. Thus, we decouple the task into two parts: (1) A scene-level **element detection task** that locates and classifies all map elements by predicting element keypoints $\mathcal{A} = \{A_i \in \mathbb{R}^{k \times 2} | i = 1, \dots, N\}$ and their class labels $\mathcal{L} = \{l_i \in \mathbb{Z} | i = 1, \dots, N\}$; (2) An object-level **sequence generation task** that produces a sequence of polyline vertices for each detected map element $(A_i, l_i)$. The definition of element keypoint representation $\mathcal{A}$ is described in § 2.2.

Correspondingly, VectorMapNet employs three modules to model these three tasks, as shown in Figure 2. (1) A BEV feature extractor that lifts sensor observations to BEV space (§ 2.1); (2) A map element detector that predicts map element keypoints $\mathcal{A}$ and class labels $\mathcal{L}$ (§ 2.2); (3) A polyline generator that completes the shapes of the HD map elements conditioned on keypoints and class labels (§ 2.3).

## 2.1 BEV FEATURE EXTRACTOR

The BEV feature extractor lifts various modality inputs into a unified feature space and aggregates these features into a canonical representation termed BEV features $\mathcal{F}_{\text{BEV}}$. We consider two common modalities: surrounding camera images $\mathcal{I}$ and LiDAR points $\mathcal{P}$.

**Camera branch.** We use ResNet to extract features from images, followed by a feature transformation module from image space to BEV space. VectorMapNet does not rely on certain feature transformation approaches and we opt to use a simple but popular variant of IPM, which produces BEV features of $\mathcal{F}_{\text{BEV}}^{\mathcal{I}} \in \mathbb{R}^{W \times H \times C_1}$. The detailed structure can be found in Appendix C.3.

**LiDAR branch.** For LiDAR data $\mathcal{P}$, we use a variant of PointPillars (Lang et al., 2019) with dynamic voxelization (Zhou et al., 2020), which divides the 3D space into multiple pillars and uses pillar-wise point clouds to learn pillar-wise feature maps. We denote this feature map in BEV as $\mathcal{F}_{\text{BEV}}^{\mathcal{P}} \in \mathbb{R}^{W \times H \times C_2}$.

For sensor fusion, we obtain the BEV features $\mathcal{F}_{\text{BEV}} \in \mathbb{R}^{W \times H \times (C_1 + C_2)}$ by concatenating $\mathcal{F}_{\text{BEV}}^{\mathcal{I}}$ and $\mathcal{F}_{\text{BEV}}^{\mathcal{P}}$, and then process the concatenated result with a two-layer convolutional network. An overview of the BEV feature extractor is shown at the bottom-left of Figure 2.

## 2.2 MAP ELEMENT DETECTOR

After obtaining BEV features, the goal of map element detector is to infer element keypoints $a_{i,j}$ from the BEV features $\mathcal{F}_{\text{BEV}}$. We leverage a variant of transformer set prediction detector (Carion et al., 2020) to achieve this goal. This detector represents map elements' locations and categories by predicting their element keypoints $\mathcal{A}$ and class labels $\mathcal{L}$.

**Keypoint representations.** In object detection problems, people use bounding box to abstract the shape of an object. Here we use $k$ key point locations $A_i = \{a_j \in \mathbb{R}^2 | j = 1, ..., k\}$, to represent the outline of a map element. However, defining keypoints for map elements is not straightforward since their are diverse. We conduct an ablation study to investigate the performance of different choices in § 3.3.

**Element queries.** The query inputs of the detector are learnable element queries $\{q_i^{\text{elem}} \in \mathbb{R}^{k \times d} | i = 1, \ldots, N_{\max}\}$, where $d$ is the hidden embedding size, and the $i$-th element query $q_i^{\text{elem}}$ is composed of $k$ keypoint embeddings $q^{\text{kp}}$: $q_i^{\text{elem}} = \{q_{i,j}^{\text{kp}} \in \mathbb{R}^d | j = 1, \ldots, k\}$.

**Architecture.** The overall architecture of the map element detector includes a transformer decoder (Vaswani et al., 2017) and a prediction head, as shown at the bottom-middle of Figure 2. The decoder transforms the element queries using multi-head self-/cross-attention mechanisms. In particular, we use the deformable attention module (Zhu et al., 2020) as the decoder's cross attention module, where each element query has a 2D location grounding. It improves interpretability and accelerates training convergence (Li et al., 2022).

The prediction head has two MLPs, which decodes element queries into element keypoints $a_{i,j} = \text{MLP}_{\text{kp}}(q_{i,j}^{\text{kp}})$ and their class labels $l_i = \text{MLP}_{\text{cls}}([q_{i,1}^{\text{kp}}, \ldots, q_{i,k}^{\text{kp}}])$, respectively. $[\cdot]$ is a concatenation operator. Each keypoint embedding $q_{i,j}^{\text{kp}}$ in the map element detector consists of two learnable parts. The first parts is a keypoint position embedding $\{e_j^{\text{kp}} \in \mathbb{R}^d | j = 1, \ldots, k\}$, indicating which position in an element keypoint the point belongs to. The second embedding $\{e_i^{\text{p}} \in \mathbb{R}^d | i = 1, \ldots, N_{\max}\}$ encodes which map element the keypoint belongs to. The keypoint embedding $q_{i,j}^{\text{kp}}$ is the addition of these two embeddings $e_i^{\text{p}} + e_j^{\text{kp}}$.

## 2.3 POLYLINE GENERATOR

Given the label and keypoints of map elements, the goal of polyline generator is to generate detailed geometrical shape of map elements. Specifically, polyline generator models a distribution $p(V_i^{\text{poly}} | a_i, l_i, \mathcal{F}_{\text{BEV}}^f)$ over the vertices of each polyline, conditioned on the initial layout (*i.e.*, element keypoints and class labels) and BEV features. To estimate this distribution, we decompose the joint distribution over $V_i^{\text{poly}}$ as a product of a series of conditional vertex coordinate distributions.

Specifically, we transform each polyline $V_i^{\text{poly}} = \{v_{i,n} \in \mathbb{R}^2 | n = 1, \ldots, N_v\}$ into a flattened sequence $\{v_{i,n}^f \in \mathbb{R} | n = 1, \ldots, 2N_v\}$ by concatenating coordinates values of polyline vertices and add an additional *End of Sequence* token ($EOS$) at the end of each sequence, and the target distribution turns into:

$$p(V_i^{\text{poly}} | a_i, l_i, \mathcal{F}_{\text{BEV}}; \theta) = \prod_{n=1}^{2N_v} p(v_{i,n}^f | v_{i,<n}^f, a_i, l_i, \mathcal{F}_{\text{BEV}}). \tag{1}$$

We model this distribution using an autoregressive network that outputs the parameters of a predictive distribution at each step for the next vertex coordinate. This predictive distribution is defined over all possible discrete vertex coordinate values and $EOS$.

**Vertices as discrete variables.** Using discrete distributions to model polyline vertices has the advantage of representing arbitrary shapes, *i.e.*, categorical distributions can easily represent various polylines, such as multi-modal, skewed, peaked, or long-tailed, that are commonly seen in our task. Thus, we quantize the coordinate values into discrete tokens and model each token with a categorical distribution. We also conduct an ablation study in Appendix D.2 to investigate other modeling choices.

**Architecture.** The autoregressive network we choose is a vanilla transformer (Vaswani et al., 2017) (see the bottom-right of Figure 2). Each polyline's keypoint coordinates and class label are tokenized and fed in as the query inputs of the transformer decoder. Then a sequence of vertex tokens are fed into the transformer iteratively, integrating BEV features with cross-attention, and decoded as polyline vertices. Note that the generator can generate all polylines in parallel.

Following PolyGen (Nash et al., 2020), we use an addition of three learned embeddings as the embedding of each vertex token: *Coordinate Embedding*, indicating whether the token represents $x$ or $y$ coordinate; *Position Embedding*, representing which vertex the token belongs to; *Value Embedding*, expressing the token's quantized coordinate value.

## 2.4 LEARNING

We train our model by minimizing the sum of map element detector loss and polyline generator loss:

$$\mathcal{L} = \mathcal{L}_{det} + \mathcal{L}_{gen} \tag{2}$$

**Map element detector loss.** Following (Wang et al., 2022; Zhu et al., 2020), the detector is trained with bipartite matching loss, thus avoiding post-processing steps like non-maximum suppression (NMS). We describe the detail of the loss function in Appendix C.4.

**Polyline generator loss.** Polyline generator is trained to maximize the log-probability of the polyline vertices. We use negative log-likelihood as its loss function:

$$\mathcal{L}_{gen} = -\frac{1}{2N_v} \sum_{n=1}^{2N_v} \log \hat{p}(v_{i,n}^f | v_{i,<n}^f, a_i, l_i, \mathcal{F}_{\text{BEV}}^f), \tag{3}$$

where $\hat{p}(v_{i,n}^f | \ldots)$ is the conditional probability of discrete coordinate value $v_{i,n}^f$, and $v_{i,<n}^f$ are ground truth discrete coordinate values with index less than $n$. The default training strategy is teacher forcing, meaning that we use ground truth keypoints as generator input. To avoid the exposure bias (Bengio et al., 2015), we further experiment with first training with teacher forcing, and then fine-tuning with predicted keypoints.

## 3 EXPERIMENTS

**Experiment protocol.** We conduct experiments on the nuScenes (Caesar et al., 2020) and Argoverse2 (Wilson et al., 2021). Following HDMapNet (Li et al., 2021), we assess the quality of a predicted HD map by comparing its components (*i.e.*, polylines) with ground truth, while the only difference is the selection of distance measure for in TP/FP matching. Both HDMapNet [1] and our paper use Chamfer distance for matching (Chamfer AP). Additionally, we also propose another distance metric termed Frechet distance (Fréchet AP), which better measures the distance between polylines by considering the order of vertices. The definitions of Chamfer AP and Fréchet AP are in § A.2. The details of dataset settings (§ A.1), implementations (§ C), and metrics (§ A.2) are presented in the Appendix as well.

Table 1: Results on nuScenes dataset. Fusion denotes the model using both images and LiDAR points as inputs. Methods with fine-tune means the model is applied two stage training strategy introduced in § 2.4

| Methods | $AP_{ped}$ | $AP_{divider}$ | $AP_{boundary}$ | mAP |
|---|---|---|---|---|
| STSU (Can et al., 2021) | 7.0 | 11.6 | 16.5 | 11.7 |
| HDMapNet (Camera) (Li et al., 2021) | 14.4 | 21.7 | 33.0 | 23.0 |
| HDMapNet (LiDAR) (Li et al., 2021) | 10.4 | 24.1 | 37.9 | 24.1 |
| HDMapNet (Fusion) (Li et al., 2021) | 16.3 | 29.6 | 46.7 | 31.0 |
| VectorMapNet (Camera) | 36.1 | 47.3 | 39.3 | 40.9 |
| VectorMapNet (Camera) + fine-tune | 42.5 | 51.4 | 44.1 | 46.0 |
| VectorMapNet (LiDAR) | 25.7 | 37.6 | 38.6 | 34.0 |
| VectorMapNet (Fusion) | 37.6 | 50.5 | 47.5 | 45.2 |
| VectorMapNet (Fusion) + fine-tune | **48.2** | **60.1** | **53.0** | **53.7** |

## 3.1 COMPARISON WITH BASELINES

**Comparison on nuScenes dataset.** We choose two closely related models, HDMapNet (Li et al., 2021) and STSU (Can et al., 2021) as our baselines. For HDMapNet, we directly take its vectorized results. STSU uses a transformer module to detect the moving objects and centerline segments. It uses an association head to piece the segments together as the road graph. In order to adapt STSU to our task, we use a two-layer MLP to predict lane segments and only keep its object branch and polyline branch. We report the average precision that uses Chamfer distance as the threshold to determine the positive matches with ground truth. $\{0.5, 1.0, 1.5\}$ are the predefined thresholds of Chamfer distance AP.

As shown in Table 1, VectorMapNet outperforms HDMapNet by a large margin under all settings (+17.9 mAP in Camera, +9.9 mAP in LiDAR, and +14.2 mAP in Fusion). Compared to camera-only and LiDAR-only, sensor fusion introduces +4.3 mAP improvement and +11.2 mAP improvement, respectively. As described in § 2.4, our two stage training strategy further boosts the performance of both camera-only and sensor fusion methods by +6.9 mAP and +8.5 mAP, respectively. STSU is -29.2 mAP lower than VectorMapNet. Since STSU treats all map elements as a set of fixed-size segments, we hypothesize that ignoring the fine geometry of map elements hurts the performance significantly.

**Results on Argoverse2.** We further compare HDMapNet and VectorMapNet on Argoverse2 dataset, shown in Table 2. Since Argoverse2 provides z-axis annotations, we give VectorMapNet results both in 2D and 3D.

In many cases of Argoverse2, the annotated boundaries and divider lines overlap with each other, making it difficult for models to separate them. It results in a drop in performance of both methods, especially in $AP_{divider}$ of HDMapNet (21.7 $AP_{divider}$ to 5.7 $AP_{divider}$) because its rasterized representation fails to handle these cases. In contrast, VectorMapNet remains competent, showing the advantage of using vectorized representation to represent overlapping elements.

Table 2: Results on Argoverse2 dataset.

| Keypoint Representaion | #dim | Fréchet Distance | | | | Chamfer Distance | | | |
|---|---|---|---|---|---|---|---|---|---|
| | | $AP_{ped}$ | $AP_{divider}$ | $AP_{boundary}$ | mAP | $AP_{ped}$ | $AP_{divider}$ | $AP_{boundary}$ | mAP |
| HDMapNet (Camera) Li et al. (2021) | 2 | - | - | - | - | 13.1 | 5.7 | 37.6 | 18.8 |
| VectorMapNet (Camera) | 2 | 43.2 | 45.5 | 52.0 | 46.9 | 38.3 | 36.1 | 39.2 | 37.9 |
| VectorMapNet (Camera) | 3 | 41.7 | 42.3 | 49.9 | 44.6 | 36.5 | 35.0 | 36.2 | 35.8 |

## 3.2 QUALITATIVE ANALYSIS

**Benefits of using polylines as primitives.** From visualizations, we find that using polylines as primitives has brought us two benefits compared with baselines: First, polylines effectively encode the detailed geometries of map elements, *e.g.* the corners of boundaries (see the red ellipses in Figure 3). Second, polyline representations prevent VectorMapNet from generating ambiguous results, as it consistently encodes direction information. In contrast, Rasterized methods are prone

to falsely generating loopy curves (see the blue ellipses in Figure 3). These ambiguities hinder safe autonomous driving. Therefore, the polyline is a desired primitive for map learning, as it can reflect real-world road layouts and explicitly encode directions.

**Benefits of posing map learning as a detection problem.** VectorMapNet works in a top-down detection manner: it models the topology of the map and the map element locations first, and then generates map element details. Visualizations show that VectorMapNet capture the map elements comprehensively, including the small elements close to edges. The high mAP of VectorMapNet over other baselines further confirms this observation. Surprisingly, Figure 4 shows that VectorMapNet can find the map elements that are not annotated in the HD map provided by the dataset.

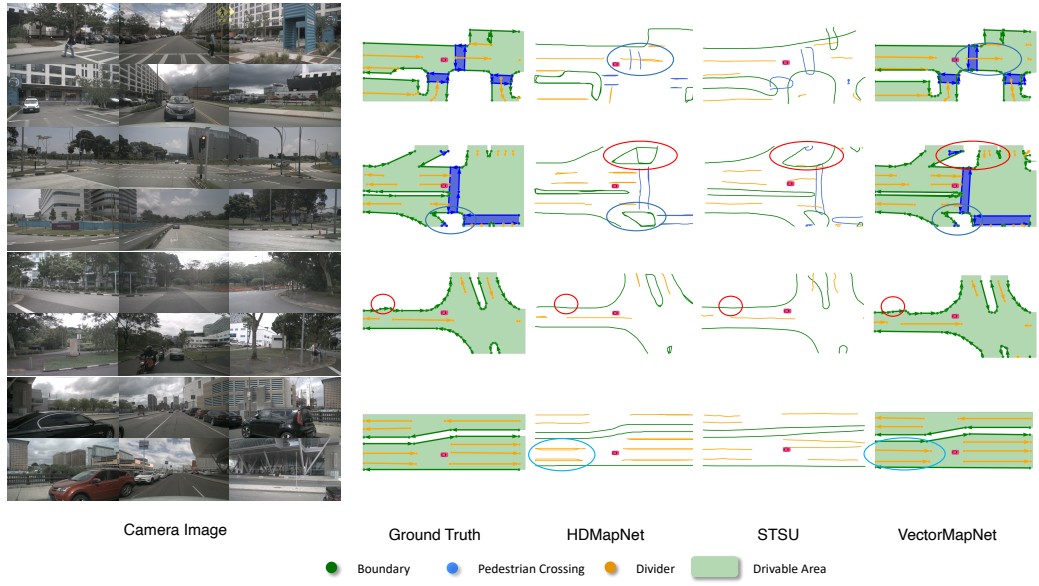

Figure 3: Qualitative results generated by VectorMapNet and baselines. We use camera images as inputs for comparisons. The areas enclosed by **red** and **blue** ellipses show that VectorMapNet can preserve sharp corners, and polyline representations prevent VectorMapNet from generating ambiguous self-looping results. Since the lack of directional information, HDMapNet and STSU cannot infer drivable areas from their predictions. It worth noting that the drivable area is inferred from several disjoint boundaries and is non-trivial to model as one object.

## 3.3 ABLATION STUDIES

We list ablation studies for keypoint representation here. For more ablation studies, please refer to Appendix D in the Appendix.

Table 3: Ablation study of keypoint representaions. $k$ is the keypoint number of each keypoint representation.

| Keypoint Representaion | $k$ | Fréchet Distance | | | | Chamfer Distance | | | |
| --- | --- | --- | --- | --- | --- | --- | --- | --- | --- |
| | | $AP_{ped}$ | $AP_{divider}$ | $AP_{boundary}$ | mAP | $AP_{ped}$ | $AP_{divider}$ | $AP_{boundary}$ | mAP |
| Bbox | 2 | **47.4** | 46.9 | **62.8** | **52.4** | **36.1** | **47.3** | **39.3** | **40.9** |
| SME | 3 | 47.0 | **47.4** | 56.9 | 50.4 | 27.6 | 34.4 | 35.4 | 32.5 |
| Extreme | 4 | 41.7 | 47.3 | 59.0 | 49.4 | 30.4 | 33.1 | 37.3 | 33.6 |

**Keypoint representations.** Since there is no straightforward keypoint design to represent map elements with few fixed number of points, we propose three simple representations as shown in Figure 5: *Bounding Box (Bbox)*, which is the smallest box enclosing a polyline, and its keypoints are defined as the top-right and bottom-left points of the box; *Start-Middle-End (SME)*, which samples the start, middle, and end point from a polyline; *Extreme Points*, which are the left-most, right-most, top-most, and bottom-most points of a polyline.

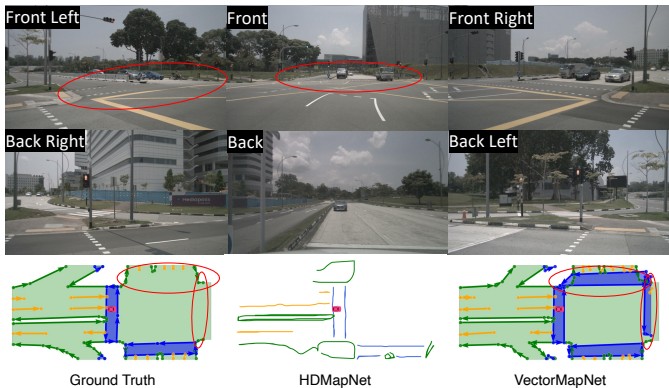

Figure 4: An example of VectorMapNet detecting unlabeled map elements. The **red ellipses** indicate two pedestrian crossings that are missing in ground truth annotations, while VectorMapNet detects it correctly. All the predictions are generated from camera images.

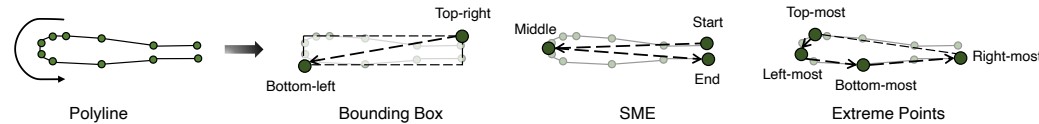

Figure 5: Three different keypoint representations are proposed here: Bounding Box (k=2), SME (k=3), and Extreme Points (k=4), where $k$ has the same definition in § 2. The arrow line indicates the direction of the example polyline, and the arrow dash lines indicate the vertices order of keypoint representations.

We experiment with these representations and list the results in Table 3. Our results show that the bounding box representation leads to the best mean average performance in both metrics, outperforming others by 2.0 Fréchet mAP and 7.3 Chamfer mAP.

## 3.4 Vectorized HD Maps for Motion Forecasting

Since predicting future motions in the complex environment heavily relies on the map information, we investigate the effectiveness of our predicted HD map in this downstream motion forecasting task.

**Task Settings.** In our setting, the motion forecasting model aims to predict a target agent's 6 plausible future trajectories (3 seconds) from past trajectories (1 second) of agents and an HD semantic map which covers an area of $60m \times 30m$. We generate data by sampling from nuScenes tracking dataset. We first retrieve agents observed in the tracking dataset and then select agents with complete 3-second future observations as the target agents. As a result, the dataset consists of 25,645 training samples and 5,460 test samples. We use three different input settings to investigate the performance of our predicted HD map: past trajectories, past trajectories with the ground truth HD map, and past trajectories with the map predicted by VectorMapNet. The motion forecaster we used is mmTransformer (Liu et al., 2021) which can optionally take vectorized maps and trajectories as inputs.

Table 4: Predicted map for motion forecasting. There are three input settings: past trajectories (denoted as Traj.), past trajectories with the human-annotated HD map from the nuScenes (denoted as Traj. + G.T. Map), and past trajectories with the predicted map from VectorMapNet (denoted as Traj. + Pred. Map). The predicted map greatly improves the prediction performance compared with the model that only use past trajectories.

| Prediction Model Inputs | minADE ↓ | minFDE↓ | MR@2m↓ |
|---|---|---|---|
| Traj. | 0.909 | 1.577 | 19.6 |
| Traj. + G.T. Map | 0.779 | 1.390 | 18.0 |
| Traj. + Pred. Map | 0.826 | 1.477 | 18.2 |

**Results.** To evaluate the performance of motion forecasting under different input settings, we report results on three commonly used metrics (Chang et al., 2019): minimum average displacement error (minADE), minimum final displacement error (minFDE) and miss rate (MR). To get the results, these metrics only account for the best trajectory out of 6 predicted trajectories. Results in Table 4 show that the map predicted by VectorMapNet has encoded environment information that greatly helps the motion forecaster, compared with the model that only takes past trajectories as inputs. The gap between the ground-truth map and the predicted map is not big either, especially in terms of MR (-0.2%). We think future research could further close the performance gap.

# 4 RELATED WORKS

**Semantic map learning.** Annotating semantic maps attracts plenty of interests thanks to autonomous driving. Recently, semantic map learning is formulated as a semantic segmentation problem (Mattyus et al., 2015) and is solved by using aerial images (Máttyus et al., 2016), LiDAR points (Yang et al., 2018), and HD panorama (Wang et al., 2016). The crowdsourcing tags (Wang et al., 2015) are used to improve the performance of fine-grained segmentation. Instead of using offline data, recent works focus on understanding BEV semantics from onboard camera images (Lu et al., 2019; Yang et al., 2021), and videos (Can et al., 2020). Only using onboard sensors as model input is particularly challenging as the inputs and target map lie in different coordinate systems. Recently, several cross-view learning approaches (Philion & Fidler, 2020; Pan et al., 2020; Li et al., 2021; Zhou & Krähenbühl, 2022; Wang et al., 2022; Chen et al., 2022) leverage the geometric structure of scenes to mitigate the mismatch between sensor inputs and BEV representations. Some methods (Casas et al., 2021; Sadat et al., 2020) use pixel-level semantic maps to solve downstream tasks, but the entire downstream pipeline needs to be redesigned to accommodate these rasterized map inputs. Beyond pixel-level semantic maps, our work extracts a consistent vectorized map around vehicles from surrounding cameras or LiDARs, which suits for existing downstream tasks like motion forecasting (Gao et al., 2020; Zhao et al., 2020; Liu et al., 2021) without modifications.

**Lane detection.** Lane detection aims to separate lane segments from road scenes precisely. Most lane detection algorithms (Pan et al., 2018; Neven et al., 2018) use a pixel-level segmentation technique combined with sophisticated post-processing. Another line of work leverages the predefined proposal to achieve high accuracy and fast inference speed. These methods typically involve handcrafted elements such as vanishing points (Lee et al., 2017), polynomial curves (Van Gansbeke et al., 2019), line segments (Li et al., 2019), and Bézier curves (Feng et al., 2022) to model proposals. In addition to using perspective view cameras as inputs, (Homayounfar et al., 2018) and (Liang et al., 2019) extract lane segments from overhead highway cameras and LiDAR imagery with a recurrent neural network. Instead of discovering the road's topology via boundaries detection, STSU (Can et al., 2021) and LaneGraphNet (Zürn et al., 2021) construct lane graphs from centerline segments that are encoded by Bézier curves and line segments, respectively. To model complex geometries in the urban environment, we leverage polylines to represent all the map elements in perceptual scopes.

**Geometric data modeling.** Another line of work closely related to VectorMapNet is geometric data generation. These methods typically treat geometric elements as a sequence, such as primitive parts of furniture (Li et al., 2017; Mo et al., 2019), states of sketch strokes (Ha & Eck, 2017), vertices of $n$-gon mesh (Nash et al., 2020) , and parameters of SVG primitives (Carlier et al., 2020). These methods generate these sequences by leveraging autoregressive models (*e.g.* Transformer). Since the directly modeling sequence is challenging for long-range centerline maps, HDMapGen (Mi et al., 2021) views the map as a two-level hierarchy. It produces a global and local graph separately with a hierarchical graph RNN. Instead of treating geometric elements as a sequence generation problem, LETR (Xu et al., 2021) models line segment as a detection problem and tackle it with a query-based detector. Unlike the above approaches that focus on single-level geometric modelings, such as scene level (*e.g.* line segments in an image) or object-level (*e.g.* furniture), VectorMapNet is designed to address both the scene level and object level geometric modeling. Specifically, VectorMapNet constructs a map by modeling the global relationship between map elements in the scene and the local geometric details inside each element.

**Learning vector representations from images** VectorMapNet bears some similarities with predicting vector graphics from raster images. In this field, several recent works (Carlier et al., 2020) and (Reddy et al., 2021) use different vector object representations to define generative models of vector images.

(Ganin et al., 2021) converts images to CAD, CanvasVAE (Yamaguchi, 2021) learns vectorized canvas layouts from images, and (Liu et al., 2022) generates vectorized stroke primitives from raster line drawing. The instance segmentation community has also been concerned with a similar task of detecting object contours in a vector form from an image. These methods (Zhang et al., 2022; Acuna et al., 2018; Liang et al., 2020) initialize a contour for every object instance and then refine the vertex positions of the contour. These methods use highly domain-dependent architectures; therefore, it would be a non-trivial task to adapt them for our task that requires detecting and generating different map elements with different semantic information and different geometry from the real-world 3D space.

## 5 Conclusions

We present VectorMapNet, an end-to-end model to tackle the HD semantic map learning problem. Unlike existing works, VectorMapNet uses polylines as the primitives to represent vectorized HD map elements. To learn these polylines, we decompose the learning problem into a detection and a generation problem. Our experiments show that VectorMapNet can generate coherent and complex geometries for urban map elements, benefiting from the polyline primitives. We believe that this novel way to learn HD maps provides a new perspective on the HD semantic map learning problem.

**Reproducibility Statement.** We detail the implementation steps and experiment settings in Appendix C.

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

## A  EXPERIMENT SETUP

### A.1  DATASET

**nuScenes** We experiment on nuScenes (Caesar et al., 2020) dataset, which contains 1000 sequences of recordings collected by autonomous driving cars. Each episode is annotated at 2Hz and contains 6 camera images and LiDAR sweeps. Our dataset setup and pre-processing steps are identical to that of HDMapNet (Li et al., 2021), which includes three categories of map elements – pedestrian crossing, divider, and road boundary – from the nuScenes dataset.

**Argoverse2** We further conduct experiments on Argoverse2 (Wilson et al., 2021) dataset. Like nuScenes, it contains 1000 logs (700, 150, 150 for training, validation and test set). Each episode provides 15s of 20Hz camera images, 10Hz LiDAR sweeps and a vectorized map. We use the same pre-processing settings as on nuScenes dataset.

### A.2  METRICS

In contrast to existing methods which generate rasterized results, our method does not require rasterizing curves on grids. Therefore, we opt not to use Intersection-Over-Union (IoU) as a metric. We use a distance-based metric to evaluate the similarity between predicted curves and ground-truth curves. We follow the instance-level evaluation metric proposed by HDMapNet (Li et al., 2021) to compare the instance-level detection performance of our model to baseline methods. The metric is average precision (AP), where positive/negative samples are based on geometric similarity, more concretely, Chamfer distance and Fréchet distance. For clarity, we call the AP based on Chamfer distance and Fréchet distance as Chamfer AP and Fréchet AP, respectively.

**Chamfer distance.** Chamfer distance is a distance measure that quantifies the similarity between two *unordered* sets. The Chamfer distance is an evaluation metric that quantifies the similarity between two unordered sets by taking into account the distance of each permutation of the elements of set as follows:

$$D_{chamfer}(\mathcal{S}_1, \mathcal{S}_2) = \frac{1}{2}(\frac{1}{|\mathcal{S}_1|} \sum_{p\in\mathcal{S}_1} \min_{q\in\mathcal{S}_2} \|p, q\|_2 + \frac{1}{|\mathcal{S}_2|} \sum_{q\in\mathcal{S}_2} \min_{p\in\mathcal{S}_1} \|q, p\|_2). \tag{4}$$

In our experiments, we use chamfer distance to calculate the distance between a prediction and a ground truth polyline set, and each polyline set is represented by uniformly sampling a polyline to $N_{pts}$ vertices, where $N_{pts}$ is set to 100 in our experiments.

**Fréchet distance.** The order of polyline vertices is not measured by Chamfer distance. Therefore, we introduce Fréchet distance as an additional measure. Fréchet distance is a measure of similarity of curves that takes both the positions and the *order* of the points along the curves into consideration. Our implementation is based on discrete Fréchet distance (Eiter & Mannila, 1994; Agarwal et al., 2014).

We use the discrete version of Fréchet distance (Eiter & Mannila, 1994; Agarwal et al., 2014) to evaluate the geometric similarity between two polyline $P$ and $Q$. We denote $\sigma(P)$ as a sequence of endpoints of the line segments of $P$. In particular, $\sigma(P) = (p_1, \ldots, p_m)$ is a sequence with $m$ vertices that uniformly sampled from the original input polyline $P$, where each position of $P$ between $p_i$ and $p_{i+1}$ can be approximated by using an affine transformation that is $p_{i+\lambda} = (1-\lambda)p_i + \lambda p_{i+1}$ and the $m$ in our experiment is set as 100.

Let $P$ and $Q$ be polyline and $\sigma(P) = (u_1, \ldots, u_p)$ and $\sigma(Q) = (v_1, \ldots, v_q)$ the corresponding sequences. A coupling $L$ is a sequence of distinct pairs between $\sigma(P)$ and $\sigma(Q)$:

$$(u_{a_1}, v_{b_1}), \ldots, (u_{a_m}, v_{b_m}). \tag{5}$$

These indexes $\{a_1, \ldots, a_m\}$ and $\{b_1, \ldots, b_m\}$ are nondecreasing surjection such that $a_1 = 1$, $a_m = p$, $b_1 = 1$, $b_m = q$ and for all $i < j \in \{1, \ldots, q\}$, $a_i \leq a_j$ and $b_i \leq b_j$.

We define the norm $\|L\|$ of the $L$ is the length of the longest pair in $L$, that is,

$$\|L\| = \max_{i=1,\ldots,m} d(u_{a_i}, v_{b_i}). \tag{6}$$

The discrete Fréchet distance between polyline $P$ and $Q$ is defined to be

$$\delta_{dF}(P, Q) = \min\{\|L\|, L \text{ is a coupling between } P \text{ and } Q\}. \tag{7}$$

This equation indicates that the distance of discrete Fréchet distance is the minimum norm of all possible couplings. To Find the coupling plausible $L$ that has the minimum norm, we use a Dynamic programming-based algorithm that is described in Algorithm 1.

---

**Algorithm 1:** The Algorithm of Discrete Fréchet Distance

---

**Input:** polyline $P = (u_1, \dots, u_p)$ and $Q = (v_1, \dots, v_q)$.
**Output:** $\delta_{dF}(P, Q)$
$ca$ : an 2d array of real with size of $(p \times q)$;
**Function** $c\,(i,\,j)$
 **if** $ca(i, j) > -1$ **then**
  **return** $ca(i, j)$;
 **else if** $i = 1$ *and* $j = 1$ **then**
  $ca(i, j) := d(u1, v1)$;
 **else if** $i > 1$ *and* $j = 1$ **then**
  $ca(i, j) := \max\{c(i - 1, 1), d(u_i, v_1)\}$;
 **else if** $i = 1$ *and* $j > 1$ **then**
  $ca(i, j) := \max\{c(1, j - 1), d(u_1, v_j)\}$;
 **else if** $i > 1$ *and* $j > 1$ **then**
  $ca(i, j) := \max\{\min(c(i - 1, j), c(i - 1, j - 1), c(i, j - 1)), d(u_i, v_j)\}$;
 **else**
  $ca(i, j) := \infty$ ;
 **end**
 **return** $ca(i, j)$;
**end**
**begin**
 **for** $i = 1$ *to* $p$ **do**
  **for** $j = 1$ *to* $q$ **do**
   ca(i, j) := -1.0;
  **end**
 **end**
 **return** $c(p, q)$;
**end**

---

## B MORE VISUALIZATIONS OF VECTORMAPNET (FUSION)

We visualized three cases of VectorMapNet (Fusion) and VectorMapNet (Camera) to demonstrate that LiDAR information can complement visual information to generate more robust map predictions. In the first case, the camera view is constrained by the nearby vehicles, so it can not provide helpful surrounding information. LiDAR sensor bypasses the nearby vehicle and provides some cue for VectorMapNet to generate a better result than its camera-only counterpart (see Figure 6). For the second case (see Figure 7), the model cannot detect the nearby parking gate because it locates in the blind zone of cameras. In contrast, the LiDAR provides depth information and helps the VectorMapNet(Fusion) detect the missing lane boundary. LiDAR points can prevent the model from falsely detecting map elements in bad weather conditions as well. As shown in Figure 8, some puddles are near the intersection. With the light reflection, these puddles visually look like a lane boundary. However, the LiDAR data shows that there does not have any bump in there. Unlike the camera-only model, this depth information from LiDAR helps our fusion model not generate a non existed lane boundary.

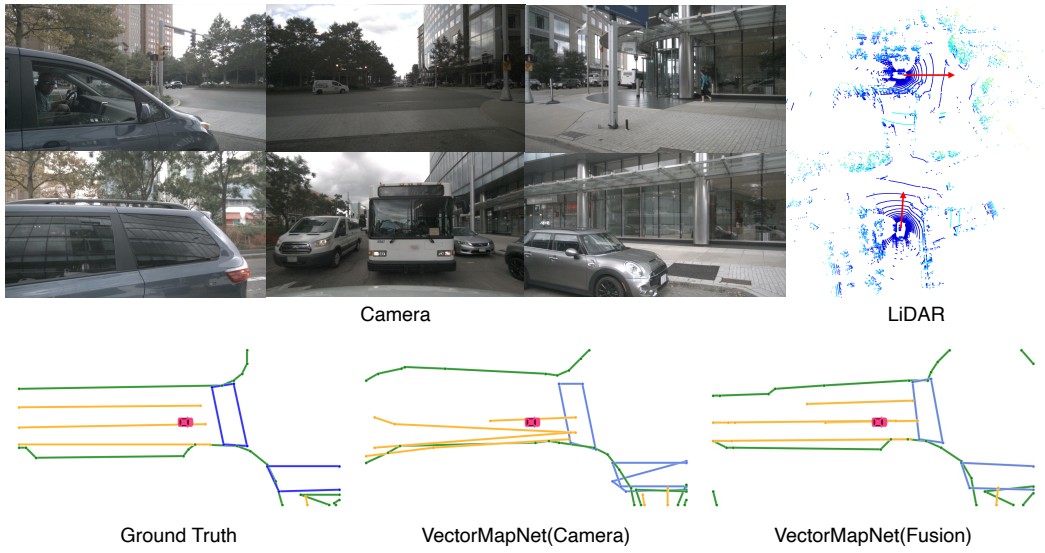

Figure 6: When the ego car cameras are occluded by the nearby vehicles, VectorMapNet(Camera) can not precept the surrounding map. With the depth cue from LiDAR, VectorMapNet(Fusion) can generate a more plausible result than its camera counterpart.

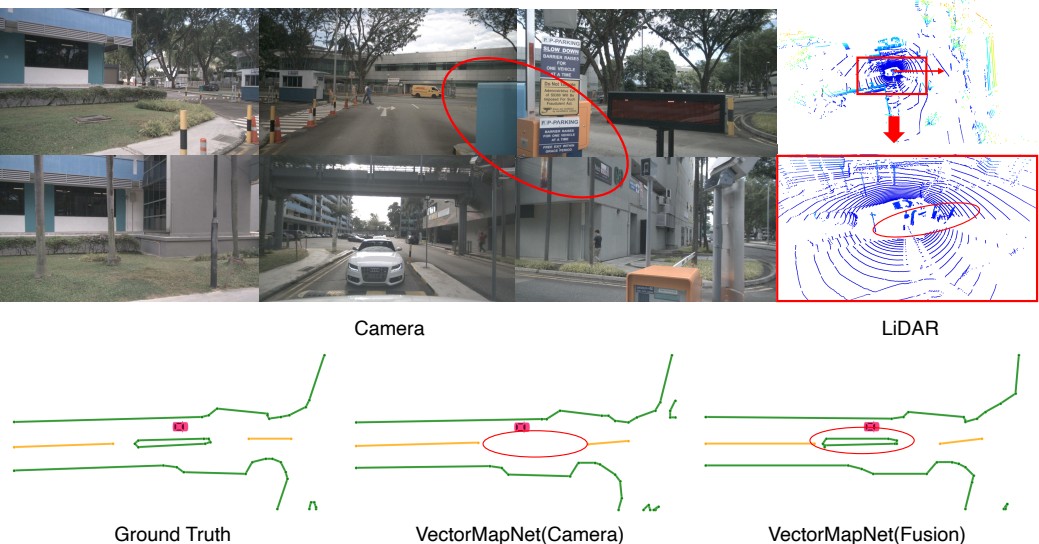

Figure 7: The blind area of onboard cameras may cause our model to miss the map elements closed ego vehicle. In contrast, we can easily find that LiDAR data has sensed some obstacles near the ego vehicle in the right-most column. With these cues, our fusion model detects the missed lane boundary by our camera-only model.

## C  IMPLEMENTATION DETAILS

### C.1  OVERALL ARCHITECTURES.

BEV feature extractor outputs a feature map with a size of $(200, 100, 128)$. It uses ResNet50 (He et al., 2016) for shared CNN backbone. We use a single layer PointNet (Qi et al., 2017) whose outputs have 64 dimensions as the LiDAR backbone to aggregate LiDAR points into a pillar. We set the number of element queries $N_{\max}$ in map element detector as 100. The transformer decoders we

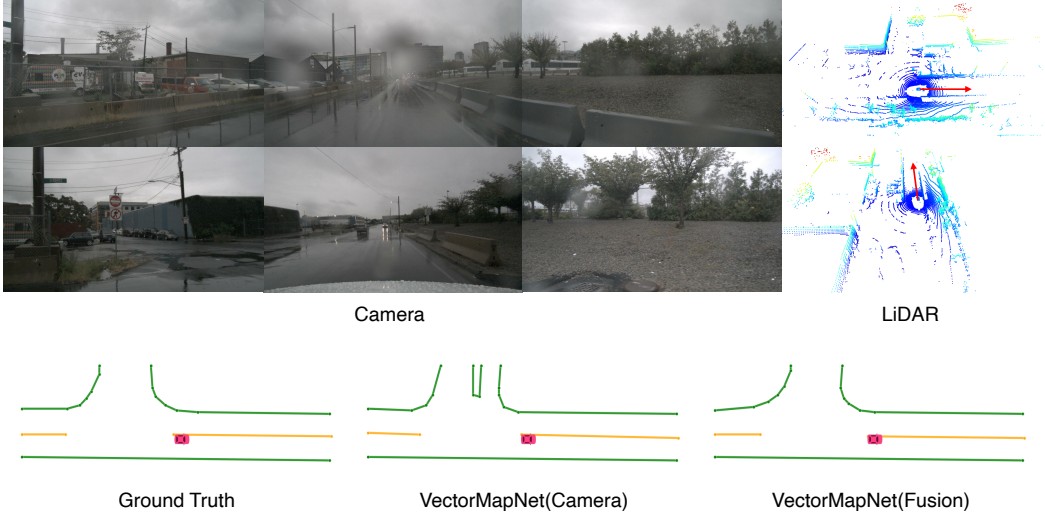

Figure 8: The qualitative results of VectorMapNet in bad weather conditions. VectorMapNet(Camera) falsely detects these puddles near the intersection as a lane boundary. The fusion result shows that the miss detection issue can be resolved by combining the depth information.

used in map element detector and polyline generator both have 6 decoder layers, and their hidden embeddings' size is 256. For the output space of polyline generator, we divide the map space (see § 2.3) evenly into $200 \times 100$ rectangular grids, and each grid has a size of $0.3m \times 0.3m$.

## C.2 TRAINING SETTINGS.

We train all our models on 8 GTX3090 GPUs for 110 epochs with a total batch size of 32. We use AdamW (Loshchilov & Hutter, 2018) optimizer with a gradient clipping norm of $5.0$. For the learning rate schedule, we use a step schedule that multiplies a learning rate by 0.1 at epoch 100 and has a linear warm-up period at the first 5000 steps. The dropout rate for all modules is 0.2, following the transformer's settings (Vaswani et al., 2017). Data augmentation is only deployed during polyline generator's training; specifically, two I.I.D. Gaussian noises are added to each input vertex's $x$ and $y$ coordinates with a probability of 0.3.

## C.3 MODEL DETAILS

**Camera Branch of Map Feature Extractor.** For image data $\mathcal{I}$, we use a shared CNN backbone to obtain each camera's image features in the camera space, then use the Inverse Perspective Mapping (IPM) (Mallot et al., 1991) technique to transform these features into BEV space. Since the depth information is missing in camera images, we follow one common approach that assumes the ground is mostly planar and transforms the images to BEV via homography. Without knowing the exact height of the ground plane, this homography is not an accurate transformation. To alleviate this issue, we transform the image features into four BEV planes with different heights ( we use $(-1m, 0m, 1m, 2m)$ in practice). The camera BEV features $\mathcal{F}_{\text{BEV}}^{\mathcal{I}} \in \mathbb{R}^{W \times H \times C_1}$ are the concatenation of these feature maps.

## C.4 LOSS

**Loss settings.** The loss function of map element detector is a linear combination of three parts: a negative log-likelihood for element keypoint classification, a smooth L1 loss, and an IoU loss for keypoints regression. The coefficients of these loss components are $2, 0.1, 1$. The matching cost of map element detector is the same as the loss combination. The loss function of polyline generator is a negative log-likelihood. We train VectorMapNet by simply summing up these losses.

**map element detector loss.** To get the loss, we first establish a correspondence between the ground-truth $(\mathcal{A}, \mathcal{L})$ and the prediction $(\hat{\mathcal{A}}, \hat{\mathcal{L}})$. Assuming the number of ground-truth map element keypoints $N$ is smaller than the number of predictions $N_{max}$, and we pad the set of ground-truth $(\mathcal{A}, \mathcal{L})$ with $\emptyset$s (no object) up to $N_{max}$. The correspondence $\sigma$ is a permutation of $N_{max}$ elements $\sigma \in \mathcal{P}$ with the lowest cost: $\sigma^* = \underset{\sigma \in \mathcal{P}}{argmin} \sum_{j=1}^{N_{max}} -\mathbb{1}_{(l_j \neq \emptyset)} \hat{p}_{\sigma(j)}(l_j) + -\mathbb{1}_{(l_j \neq \emptyset)} \mathcal{L}_{keypoint}(a_j, \hat{a}_{\sigma(j)})$, where $\hat{p}_{\sigma(j)}(l_j)$ is the probability of class label $l_j$ for the prediction with index $\sigma(j)$, and the loss of keypoints parameters $\mathcal{L}_{keypoint}$ is an addition of a smooth L1 loss and an IoU loss. With these notations we define the loss of detector as:

$$\mathcal{L}_{det} = \sum_{j=1}^{N_{max}} - \log \hat{p}_{\sigma^*(j)}(l_j) + \mathbb{1}_{(l_j \neq \emptyset)} \mathcal{L}_{keypoint}(a_j, \hat{a}_{\sigma^*(j)}),$$

where $\sigma^*$ is the optimal assignment computed by Hungarian algorithm (Kuhn, 1955).

## D  MORE ABLATION STUDIES

### D.1  CURVE SAMPLING STRATEGIES

Table 5: Ablation study of curves sampling strategies.

| | Fréchet Distance | | | | Chamfer Distance | | | |
|---|---|---|---|---|---|---|---|---|
| Vertex Sampling Method | $AP_{ped}$ | $AP_{divider}$ | $AP_{boundary}$ | mAP | $AP_{ped}$ | $AP_{divider}$ | $AP_{boundary}$ | mAP |
| curvature-based | 47.0 | 47.4 | 56.9 | 50.4 | 27.6 | 34.4 | 35.4 | 32.5 |
| fixed interval | 26.0 | 23.6 | 37.1 | 28.9 | 14.6 | 17.6 | 18.7 | 17.0 |

We use two approaches to sample polylines. The first is based on the original nuScenes setting (Caesar et al., 2020), which samples vertices at the position where the curvature changes are beyond a certain threshold. The second is to sample the vertices at fixed intervals ($1m$). We compare our methods under these two sampling strategies and the results are shown in Table 5. The curvature-based sampling outperforms its fixed-sampling counterpart by a large margin and achieves a leading 21.5 Fréchet mAP and 15.5 Chamfer mAP. We hypothesize that the fixed-sampling method involves a large set of redundant vertices that have negligible contributions to the geometry, thus under-weighs the essential vertices (*e.g.* the vertices at the corner of a polyline) in the learning process.

### D.2  VERTEX MODELING METHODS.

Table 6: Ablation study of vertex modeling methods.

| | Fréchet Distance | | | | Chamfer Distance | | | |
|---|---|---|---|---|---|---|---|---|
| Modeling Method | $AP_{ped}$ | $AP_{divider}$ | $AP_{boundary}$ | mAP | $AP_{ped}$ | $AP_{divider}$ | $AP_{boundary}$ | mAP |
| discrete | 47.0 | 47.4 | 56.9 | 50.4 | 27.6 | 34.4 | 35.4 | 32.5 |
| continuous | 38.0 | 41.6 | 46.1 | 41.9 | 26.5 | 28.1 | 30.1 | 26.5 |

We investigate both discrete and continuous ways to model polyline vertices. The discrete version of polyline generator is described in § 2.3. With the same model structure, we follow SketchRNN (Ha & Eck, 2017) and use mixture of Gaussian distributions to model the vertices of polylines as continuous variables. The comparison is shown in Table 6. We find that using discrete embeddings vertex coordinates results in a considerable gain in performance, with Chamfer mAP increasing from 18.2 to 32.5 and the Fréchet mAP increasing from 26.8 to 50.4. These improvements suggest that the non-local characteristic of categorical distribution helps our model to capture complex vertex coordinate distributions.

