# OpenReview forum: "VectorMapNet: End-to-end Vectorized HD Map Learning"
_ICLR.cc/2023/Conference — Submitted to ICLR 2023_

### Official Review · Reviewer_oajd · 2022-10-20

**Confidence:** 5
**Correctness:** 2
**Technical Novelty And Significance:** 3
**Empirical Novelty And Significance:** 3
**Recommendation:** 3

**Clarity, Quality, Novelty And Reproducibility:**

The proposed algorithm is generally clear presented.
But the novelty is limited, some previous vector extraction methods are missing for comparison. For example,  E2EC for contour extraction, PolyWorld for vector extraction and the Enhanced-iCurb et al. Those end-to-end approaches also conduct vector extraction algorithm and perform well on many benchmarks and can be directly applied to HD map generation.
The source code is not provided and some details are missing, e.g., the length of polyline for training, which weaken the reproducibility.

Reference
#1 Zhang T, Wei S, Ji S. E2EC: An End-to-End Contour-based Method for High-Quality High-Speed Instance Segmentation. InProceedings of the IEEE/CVF Conference on Computer Vision and Pattern Recognition 2022 (pp. 4443-4452).

#2 Zorzi S, Bazrafkan S, Habenschuss S, Fraundorfer F. PolyWorld: Polygonal Building Extraction with Graph Neural Networks in Satellite Images. InProceedings of the IEEE/CVF Conference on Computer Vision and Pattern Recognition 2022 (pp. 1848-1857).

#3 Xu Z, Liu Y, Gan L, Hu X, Sun Y, Liu M, Wang L. csBoundary: City-Scale Road-Boundary Detection in Aerial Images for High-Definition Maps. IEEE Robotics and Automation Letters. 2022 Feb 24;7(2):5063-70.

**Details Of Ethics Concerns:**

There is no ethics concerns.

**Strength And Weaknesses:**

The proposed end-to-end vector extraction is interesting. However, some important discussion is missing, which is listed as below:

1. In the ablation study, bounding box representation is stated as the optimal representation. Why not represented the polyline extraction as the instance segmentation (with boundary sequence encoding) ? The reviewer cannot figure out the advantage to instance segmentation.

2. How to define the length of polyline? In the polyline generator, each element keypoints is reserved as the tokens and a lookup table is built with addition EOS. Since different polyline has different length on HD map, how to determine the minimum number of keypoints for each polyline? The reviewer notice that the author discuss the approaches to sample polylines in D.1 But the minimum number of keypoints for different type of polylines are not found.

3. The author only evaluated the performance with mAP metric. How about other metrics, such as boundary mIoU, top-k accuracy (for classification branch)?


**Summary Of The Paper:**

This paper present an end to end vectorization algorithm for generating HD map. It takes the multi-view images and LiDAR point cloud as the input and produce HD map via series keypoints. At the first stage, a BEV features are extracted from onboard sensor data. Then element keypoints are encoded via map element detector and a polyline generator is utilized to encode the polylines. Experiments on two datasets demonstrate the effectiveness of the approach.

**Summary Of The Review:**

The topic of end-to-end vector extraction is promising. Combined with LiDAR and multi-view images, HD map can be smoothly generated. However, the main concern is not solved in the paper (see weakness parts). In addition, several related works are not presented for fair comparisons of the HD maps.
The reviewer  lean to reject the current form if the above concerns are not clearly presented.

---

> ### Author Response · Authors · 2022-11-16
> **Response to Reviewer oajd - part 1**
>
> We thank Reviewer oajd for reviewing our paper and providing helpful comments on our work. Below please see our detailed responses.
>
> ## Q1. Why not represent the polyline extraction as the instance segmentation (with boundary sequence encoding) ?
>
> > In the ablation study, bounding box representation is stated as the optimal representation. Why not represent the polyline extraction as the instance segmentation (with boundary sequence encoding) ? The reviewer cannot figure out the advantage to instance segmentation.
>
> Thanks for your feedback. We would like to respectfully clarify the following two aspects:
> 1. The definition of keypoints is different from vertices. The keypoints are just the intermediate representations of VectorMapNet that are predicted by the map element detector. The purpose of it is to locate and sketch the location and geometry of polylines. And its results are passed to the polyline generator. We use polylines as output representations.
> 2. Contour-based instance segmentation methods do not easily apply. We had a similar idea when working on this project. But we found that there is no straightforward way to adopt these methods to the vector map learning task for two reasons:
>     1. They represent instances in a close form. Contours can not represent all the types of map elements. For example, lane dividers and road boundaries are usually represented as open curves, road signs are represented as points. In addition, the drivable area is inferred from several disjoint boundaries and is non-trivial to model as one object.
>     2. They ignore the direction of the vectors. The direction is essential for the autonomous driving system and is one of the reasons we want to develop a vector map learning method. Downstream tasks such as trajectory prediction require the directions of the divider and boundary to infer the possible location of the nearby agents.
>
> We have included this analysis and cited the paper you mentioned in our revised related works.
>
> On the other hand, changing the model of the polyline generator from a transformer to a contour-based instance segmentation method like E2EC[1] will not alter our VectorMapNet frameworks. It is just a specific choice of implementation.
>
> [1] Zhang, Tao, Shiqing Wei, and Shunping Ji. "E2EC: An End-to-End Contour-based Method for High-Quality High-Speed Instance Segmentation." Proceedings of the IEEE/CVF Conference on Computer Vision and Pattern Recognition. 2022.
>
> ## Q2
> >How to define the length of polyline? In the polyline generator, each element keypoints is reserved as the tokens and a lookup table is built with addition EOS. Since different polylines have different lengths on HD maps, how to determine the minimum number of keypoints for each polyline? The reviewer notices that the author discusses the approaches to sampling polylines in D.1 , but the minimum number of keypoints for different types of polylines are not found.
>
> ### Q2.1 How to define the length of polyline?
>
> We want to clarify the definition of keypoints and vertices. In D.1, we discussed the sampling strategies of polyline vertices instead of the keypoints, and these strategies determine the length of polylines. In principle, a polyline can be as short as one vertex while representing a point, and it can also be as long as 42 while representing a curve lane. In practice, we selected the default nuScenes sampling strategy in our experiments, which converts vector HD maps to polylines by applying the Douglas-Peucker algorithm[1]. We have plotted the polyline vertex number distribution of the nuScenes validation set, which is included in the supplementary material.
>
> [1] Saalfeld, Alan. "Topologically consistent line simplification with the Douglas-Peucker algorithm." Cartography and Geographic Information Science 26.1 (1999): 7-18.
>
> ### Q2.2 How to determine the minimum number of keypoints for each polyline?
> As mentioned in the last answer, the keypoints are intermediate features. The numbers of keypoints for different types of polylines are fixed and unchanged once we select a keypoint representation. In Figure 5 and section 3.3, we have described how to determine the keypoint locations for each keypoint representation given a polyline.
>
> The minimum keypoint number for each representation is the same as their definition. Specifically, the bounding box is 2, the SME is 3, and the extreme point is 4.

---

> > ### Author Response · Authors · 2022-11-16
> > **Response to Reviewer oajd - part 2**
> >
> > ## Q3. Novelty
> > > The novelty is limited, some previous vector extraction methods are missing for comparison. For example, E2EC for contour extraction, PolyWorld for vector extraction and the Enhanced-iCurb et al. Those end-to-end approaches also conduct vector extraction algorithm and perform well on many benchmarks and can be directly applied to HD map generation
> > >
> > >[1] E2EC: An End-to-End Contour-based Method for High-Quality High-Speed Instance Segmentation
> > >
> > > [2] PolyWorld: Polygonal Building Extraction with Graph Neural Networks in Satellite Images.
> > >
> > >[3] csBoundary: City-Scale Road-Boundary Detection in Aerial Images for High-Definition Maps.
> >
> > Thanks for your suggestion. We agree that these works are promising in their task settings. However, there is no straightforward way to adapt them to our online map learning tasks for three general reasons:
> > 1. **These methods' settings are very different from Vector Map learning.** While previous works learn from an aerial image where the input and output perfectly align, our task is more challenging. In particular, there are **three challenges**:
> >
> >     1. The map learning task involves perspective-to-BEV view transformation; not all map elements are fully visible from the sensors. In extreme cases, some map elements even are occluded by vehicles.
> >     2. The geometrical and topological relationships between map elements are complex. Unlike building extraction or boundary detection, the map elements may overlap, and traffic rules can define the road boundary. For example, two traffic cones connected with a wire may indicate the road boundary.
> >     3. The map elements for autonomous driving have more complex geometric structures. We list more details at the end of this rebuttal.
> >
> >     **We are the first method that tackles these three challenges with an end-to-end model.** In particular, we first chose polylines to model the heterogeneous nature of map elements. And then, we view the polyline map construction as a detection problem and use a query-based detector to model the relation between map elements and handle the overlapping between them. Finally, we use an autoregressive model to encode the map element's direction and the local geometries.
> >
> > 2. **These representations cannot represent all map elements in our task.** HD maps are typically composed of a mixture of different geometries, such as points, lines, curves, and polygons. So a plausible vector representation requires representing different geometry. Besides, this representation needs to consider the usability for downstream tasks.
> >
> > 3. **It is non-trivial to adopt these methods to model polylines.** The polylines have variable lengths and are encoded with orders. So the methods should be able to generate sequential results with varying lengths.
> > Below, we give a specific response for each line of work to clarify why we didn't compare these methods with VectorMapNet.
> >
> > **For contour-based Instance segmentation methods (e.g., E2EC[2]):**
> >
> > The reason is the same as the answer to part 2 of question 1.
> >
> > **For building extraction methods(e.g.,PolyWolrd and BuildMapper[1]):**
> >
> > These methods work similarly to the contour-based image segmentation methods. For example, a recently proposed building extraction method[1] claims to be an extension of E2EC[2]. And these methods also assume the result is a closed polygon and only model one type of geometry(i.e., buildings). So they have similar problems with contour-based Instance segmentation methods when adapting to the map learning task.
> >
> > **For aerial boundary detection methods (e.g., csBoundary[3]):**
> >
> > Aerial boundary detection methods are typically used for SD map construction. This line of work focuses on discovering the topology of maps, such as road networks, and does not model lane-level information that is important for autonomous driving. Besides, these methods usually use graph-based modeling( e.g., csBoundary), which is hard to adapt to model local geometries like curvy curbs and is redundant for representing map elements as there are no vertices of polylines that have degrees larger than 2. Besides, vehicles must distinguish different map elements from the environment in autonomous driving to navigate the world. But the aerial boundary detection methods are designed to model the single class (boundary), ignore the instance information inside a map, and treat road boundaries as a whole. So these works cannot fulfill our vector map learning requirements that models need to distinguish map elements and classify them into different classes.
> >
> > [1] Shiqing, Wei, et al. "BuildMapper: A Fully Learnable Framework for Vectorized Building Contour Extraction." arXiv preprint arXiv:2211.03373 (2022)
> >
> > [2] Zhang, Tao, Shiqing Wei, and Shunping Ji. "E2EC: An End-to-End Contour-based Method for High-Quality High-Speed Instance Segmentation."
> >
> > [3] Xu, Zhenhua, et al. "csBoundary: City-Scale Road-Boundary Detection in Aerial Images for High-Definition Maps."

---

> > > ### Author Response · Authors · 2022-11-16
> > > **Response to Reviewer oajd - part 3**
> > >
> > > ## Q4. Why use mAP as metric.
> > > > The author only evaluated the performance with mAP metric. How about other metrics, such as boundary mIoU, top-k accuracy (for classification branch)?
> > >
> > > As we mentioned in the general response, our metric follows the HDMapNet[1], specifically designed based on the characteristic of the polylines.
> > > The boundary IoU is a metric that measures the similarity between two semantic regions, and it still relies on the mAP metric to evaluate the instance segmentation results. However, we cannot use boundary IoU to evaluate polylines because we cannot represent each polyline as a region. Moreover, the chamfer distance in our evaluation pipeline has considered the similarity between the two polylines.
> > >
> > > As for the top-K accuracy, we only have a limited number of categories (3 in our case). Therefore, it doesn't make sense to evaluate top-K accuracy. We will include top-K accuracy for other datasets in the future.
> > >
> > > We have also considered the similarity of two polylines with direction by using the Fréchet distance. Thus, we think our evaluation metric is sufficient to reflect the performance of our model.
> > >
> > > ## Q5 Reproducibility
> > > > The source code is not provided and some details are missing, e.g., the length of polyline for training, which weakens the reproducibility.
> > >
> > > To ensure reproducibility, we upload our code in the revised supplementary material.
> > >
> > > If we have successfully addressed your issues and concerns, we would strongly appreciate an increased score. Otherwise, please let us know and we are happy to provide additional experiments and/or discussion to allay your issues and concerns.

---

### Official Review · Reviewer_5FhD · 2022-10-24

**Confidence:** 3
**Correctness:** 4
**Technical Novelty And Significance:** 4
**Empirical Novelty And Significance:** 4
**Recommendation:** 8

**Clarity, Quality, Novelty And Reproducibility:**

The proposed idea is original. The authors concisely describe the details of the method and the improvement from the existing methods.

**Strength And Weaknesses:**

Strengths
+ The map elements are represented as a set of polylines and this can express various map components such as road boundary, pedestrian crossing and stop line.
+ The keypoint representation is investigated in ablation study and determined.
+ The performance is evaluated on public dataset and the proposed method achieves higher accuracy than HDMapNet.
+ The polyline generator generates the order of polyline vertices and these are critical information for HDMap.

Weaknesses
- There are few discussions regarding the issues below.
- How about processing time?
- Are multi-camera input mandatory?  Can a front camera output HDMap in front of the vehicle?
- How about the sensitivity of VectorMapNet to camera position, intrinsic parameters, extrinsic parameters?


**Summary Of The Paper:**

The authors propose the generation algorithm of vectorized HD map from onboard sensors such as cameras and LiDARs.  They represent map elements as a set of polylines and the positions are learned from extracted BEV features.  After element keypoints are detected, the trained decoder outputs polylines of every elements.  The experimental study shows that their proposed VectorMapNet achieves state-of-the-art performance on the public nuScenes and Argoverse2 dataset.

**Summary Of The Review:**

The proposed method is novel and achieves definitely higher accuracy than the existing methods.  I think that this paper deserves acceptance to ICLR.

---

> ### Author Response · Authors · 2022-11-16
> **Response to Reviewer 5FhD**
>
> We thank reviewer 5FhD for the positive comments and for providing thoughtful feedback on our work. Below please see our detailed responses.
>
> ## Q1. How about processing time?
>
> Following the suggestion, we have conducted an experiment to compare the processing time of HDMapNet and VectorMapNet in terms of FPS. All experiments are run on the same device (1 Intel Xeon Platinum 8358 CPU @ 2.60GHz + 1 RTX 3090). We believe that in practical applications, with additional optimization, VectorMapNet is able to run in real-time (typically 10 FPS for self-driving) on an advanced onboard GPU, like Orin.
>
> | Model        | Frame per second (FPS) | Speedup |
> |--------------|------------------------|---------|
> | HDMapNet     | 0.5                    | 1x      |
> | VectorMapNet | 8.3                    | 16.6x   |
>
> ## Q2.1 Is multi-camera input mandatory? Can a front camera output HDMap in front of the vehicle?
>
> Multi-camera input is not mandatory, and it mainly helps with view consistency. VectorMapNet can work with a front camera and output the map in the front of the vehicle.
>
> Additionally, VectorMapNet can handle arbitrary numbers of camera inputs. As shown in the experiment section, our model can produce competitive results in both Argoverse2 and nuScenes datasets, which contain images from 7 cameras and images from 6 cameras for samples from these datasets respectively.
>
> But evaluation is tricky, so we did not do it and left this to future work.
> VectorMapNet is not restricting its feature extraction backbone, so it can handle the various inputs by converting the backbone to the model designed for this type of model.
>
> ## Q3. How about the sensitivity of VectorMapNet to camera position, intrinsic parameters, extrinsic parameters?
>
> Thanks for this suggestion. To probe the robustness of VectorMapNet, we follow lift-splat-shoot [1], which tests the model under the noise that occurs in self-driving, such as camera extrinsic being biased.
>
> The table shows that training the model with noisy extrinsic can lead to better test-time performance. And our model maintains its good performance for high amounts of extrinsic noise. The results show our model's robustness against extrinsic noise.
>
> | mAP                        | Test time extrinsic noise |       |      |      |
> |----------------------------|---------------------------|-------|------|------|
> | Train time extrinsic noise | 0                         | 0.1   | 0.3  | 0.6  |
> | 0                          | 42.2                      | 42.6  | 42.4 | 42.5 |
> | 0.1                        | **43.8**                      | 43.6  | 43.5 | 43.6 |
> | 0.3                        | 43.0                      | 43.0  | 43.1 | 43.1 |
> | 0.6                        | 42.6                      | 42.72 | 42.8 | 42.9 |
>
> [1] Philion, Jonah, and Sanja Fidler. "Lift, splat, shoot: Encoding images from arbitrary camera rigs by implicitly unprojecting to 3d." European Conference on Computer Vision. Springer, Cham, 2020.
>
> We thank you for your review of our manuscript, and we hope that the above responses adequately address all concerns.

---

### Official Review · Reviewer_sk1h · 2022-10-24

**Confidence:** 3
**Correctness:** 4
**Technical Novelty And Significance:** 3
**Empirical Novelty And Significance:** 3
**Recommendation:** 5

**Clarity, Quality, Novelty And Reproducibility:**

* Quality is high - experiments are well-conducted and ablations are provided.
* Clarity is low.
* Originality is moderate.
* Reproducibility is low - the system is complicated and the code is not provided.

**Strength And Weaknesses:**

\+ The paper addresses an important problem in self-driving.

\+ Experimental results show improvement over baselines.

\- The paper is quite difficult to read.

\- It is not fully clear how established the experimental protocol is (see below).

Questions:
* It is not clear from the beginning what kind of map is proposed by this paper. In particular, term polyline requires an upfront explanation. Also, it is not immediately clear what polylines represent and how to use them for autonomous driving. Finally, how is coordinate system handled, does the network place the origin at the origin of the LIDAR map?
* It is not clear how the metrics are calculated. Looking at the Figure 3, it seems like automatic comparison of the map to the ground truth is a challenging problem. Was the evaluation protocol used by this paper established in prior work (pointers?) or is it one of the contributions of this paper as well?

**Summary Of The Paper:**

The paper proposes a learned system to generate semantic 2D maps for driving based on RGB on-vehicle images and LIDAR point cloud. The novelty aspect is in using polylines to represent the map, which also influences the choice of the architecture (the paper follows Nash et al. which previously addressed CAD model generation). The experimental results show improvement over baselines.

**Summary Of The Review:**

I am learning towards rejection due to low clarity.

---

> ### Author Response · Authors · 2022-11-16
> **Respond to Reviewer sk1h**
>
> We thank Reviewer sk1h for the reviews and questions. Please see our detailed responses below.
>
> ## Q1.1 It is not clear from the beginning what kind of map is proposed by this paper.
> High-definition semantic map is a widely used map representation in the autonomous driving field. Different from a LiDAR map, a semantic map is often (traditionally) manually labeled. In this work, we follow the problem setting and HD map formats proposed in HDMapNet [1], where many other works also follow. For example, a recent work MapTR [2] acknowledges our contribution and follows our setting. They use the same HD map formats and the same problem setting. Moreover, they use the same polyline definition and follow our evaluation protocol.
> Concretely, our task is to detect every map element instance in the scene with its geometry and semantic label. The geometry and location of each map instance is represented as a polyline with a sequence of vertices. We have provided an example of expected polyline map outputs in supplementary material.
>
> [1] HDMapNet: An Online HD Map Construction and Evaluation Framework. arXiv:2107.06307
>
> [2] MapTR: Structured Modeling and Learning for Online Vectorized HD Map Construction. arXiv:2208.14437
>
> ## Q1.2 It is not immediately clear what polylines represent and how to use them for autonomous driving.
>
> The polylines can encode the structured information of HD maps, such as the road's direction, and many methods can efficiently process it (e.g., Pointnet ++[1]). Most autonomous driving datasets, such as the nuScenes and the Argoverse2 dataset, use the polyline to represent HD maps. And many downstream tasks are also designed to take polyline maps as input. For example, one of the most popular map encoding methods, VectorNet[2], takes polylines as input for motion forecasting. In Section 3.4, we conduct a downstream experiment and show that our predicted polyline map greatly helps the motion forecasting task.
>
> [1] Qi, Charles Ruizhongtai, et al. "Pointnet++: Deep hierarchical feature learning on point sets in a metric space." Advances in neural information processing systems 30 (2017).
>
> [2] Gao, Jiyang, et al. "Vectornet: Encoding hd maps and agent dynamics from vectorized representation." Proceedings of the IEEE/CVF Conference on Computer Vision and Pattern Recognition. 2020.
>
> ## Q1.3 Finally, how is coordinate system handled, does the network place the origin at the origin of the LIDAR map?
> As for the coordinate system, we place ego vehicle as the origin. Then we set x-axis towards the front, y-axis towards the left and z-axis towards the top of ego vehicle.
>
> Different from traditional mapping tasks whose goal is to build a global LiDAR/semantic map, where a global coordinate needs to be determined. In our task, the goal is to understand the local semantic map around the vehicle, therefore using ego-vehicle coordinate is enough for local motion prediction and planning.
>
> ## Q2. Evaluation protocol？
> >It is not clear how the metrics are calculated. Looking at the Figure 3, it seems like automatic comparison of the map to the ground truth is a challenging problem.
> >
> >Was the evaluation protocol used by this paper established in prior work (pointers?) or is it one of the contributions of this paper as well?
>
> Our evaluation protocol largely follows the one proposed in HDMapNet [1], which is average precision (AP). This is similar to the metric in object detection tasks while the only difference is the selection of distance measure for in TP/FP matching. Both HDMapNet [1] and our paper use Chamfer distance for matching. Additionally, we also propose another distance metric termed Frechet distance, which better measures the distance between polylines by considering the order of vertices. More details are discussed in section A.2. When comparing with other methods, we use Chamfer distance-based AP. When doing ablation studies of our model, we provide both results of Chamfer distance based AP and Frechet distance-based AP.
>
> [1] HDMapNet: An Online HD Map Construction and Evaluation Framework. arXiv:2107.06307
>
> ## Q3. Reproducibility is low - the system is complicated and the code is not provided.
> To facilitate reproducibility, we upload our code in the revised supplementary material.
>
> If we have successfully addressed your concerns, we would strongly appreciate an increased score. Otherwise, please let us know and we are happy to provide additional experiments and/or discussion to allay your concerns.

---

> > ### Comment · Reviewer_sk1h · 2022-11-22
> > **Response from reviewer**
> >
> > Thank you for the explanations and uploading the code.
> >
> > **"For example, a recent work MapTR [2] acknowledges our contribution and follows our setting. They use the same HD map formats and the same problem setting.
> > [2] MapTR: Structured Modeling and Learning for Online Vectorized HD Map Construction. arXiv:2208.14437"**
> >
> > Pointing to a deanonymized version of your work (found in the referred paper) breaks double-blind process. This gives more ground for rejection, unfortunately.
> >
> > Beyond anonymity, I still think the clarify of the manuscript remains low. As an example, this paragraph is hard to parse due to peculiar capitalization and use of semi-colon: "These map elements include but are not limited to : Road boundaries, boundaries of roads that split roads and sidewalks. Typically, they are curves with irregular shapes and arbitrary lengths; Lane dividers, boundaries of the lanes in the road. Usually they are straight lines; Pedestrian crossings, regions with white markings where pedestrians can legally cross the road. Usually they are quadrilaterals."
> >
> > Moreover, the current uploaded version exceeds the page limit of 9 pages.
> >
> > I am keeping my rating unchanged, and still leaning towards rejection.

---

### Official Review · Reviewer_TD4k · 2022-10-30

**Confidence:** 3
**Correctness:** 3
**Technical Novelty And Significance:** 3
**Empirical Novelty And Significance:** 3
**Recommendation:** 6

**Clarity, Quality, Novelty And Reproducibility:**

The paper is fairly well-written and contains neat illustrations. It also contains enough detail for someone with sufficient expertise to implement and reproduce the results.

**Strength And Weaknesses:**

Strengths

1. The proposed method is able to automatically generate vector maps without human input. This is significant as the process of creating vector maps from sensor data usually requires significant manual effort and/or are not able to generalize to new types of sensor data.

2. Nearly all commercial mapping software today uses vector representations and the proposed method can therefore naturally interface with existing map ecosystems. The output vector maps are more compact and easier to interpret than those generated by other methods which work with rasterized grids.

3. The use of transformers for predicting vector elements seems well suited to the problem as vector nodes naturally map to the “tokens” used by transformers in other domains.

4. The results are quite strong and the vector maps generated by the proposed approach are more accurate than those generated by existing methods. Specifically, the approach significantly outperforms HDMapNet (the closest existing method) on nuScenes and ArgoVerse in terms of average precision (AP).

Weaknesses

1. The related work described in the paper seems quite limited. Instead of only focussing on HDMapNet, there is a whole category of research [eg. 1] that deals with converting raster images to vector images which are closely related and should be included. Have transformers been used before for vectorization?

2. The proposed approach works only with a single-pass mapping of a particular area. How would multiple mapping sessions be integrated to obtain higher-quality maps?

3. There is no mention of the computational requirements and runtime of the approach. Does it work in real-time? Can it be run on an edge node or does it have to run on more powerful machines in the cloud?

[1] Carlier, Alexandre, et al. "Deepsvg: A hierarchical generative network for vector graphics animation." Advances in Neural Information Processing Systems (2020)

**Summary Of The Paper:**

This paper introduces a method for extracting compact and interpretable vector maps from input camera and lidar data. The method consists of three key components: a feature extractor, a map element predictor and a polyline generator. The feature extractor uses a CNN to extract features from the images and the point cloud. The map element predictor uses a transformer to identify and localize road boundaries, dividers and pedestrian crossings which are then described as keypoints. The polyline generator then uses another transformer-based model to connect adjacent keypoints into complete vector shapes. While this is not the first method to construct vector maps from raw sensor data, it performs significantly better than existing approaches such HDMapNet.

**Summary Of The Review:**

Overall, the paper makes a good contribution. While the task of predicting vector maps from raw sensor data is not new, the transformer-based approach is interesting and achieves a significant increase in performance. I would recommend the related work be updated and the questions above addressed.

---

> ### Author Response · Authors · 2022-11-16
> **Response to Reviewer TD4k**
>
>
> Dear Reviewer TD4k,
>
> Thank you for the detailed comments and insightful questions.
> # Q1.1 The related work described in the paper seems quite limited.
> > The related work described in the paper seems quite limited. Instead of only focussing on HDMapNet, there is a whole category of research [eg. 1] that deals with converting raster images to vector images which are closely related and should be included. Have transformers been used before for vectorization?
>
> Thanks for pointing this out! We agree that there are some previous works that extract vector graphics from raster images. We have included DeepSVG[1] and other methods[2,3,4] that learn vector representations from images, in the revised manuscript.
>
> # Q1.2 Have transformers been used before for vectorization?
> There are a couple of recent works that propose transformer models for vectorization . For example, Ganin et. al. [1] converts images to CAD, CanvasVAE[4] learns vectorized canvas layouts from images, and Liu et. al.[2] generates vectorized stroke primitives from raster line drawing.
>
> Our map learning task is different from these tasks in two ways: (1) map elements lie in the 3D space, which requires learning view transformations; (2) it requires reconstruction of map elements of various semantics and shapes. For example, in HD semantic maps, lanes are usually represented as curves, pedestrian crossings are often represented as polygons, stop signs and traffic lights are represented as points. The heterogeneous nature of map elements calls for a unified vectorized representation, and we find polylines to be an effective choice. We have included more qualitative results in supplementary results, including centerline prediction .
>
> [1] Carlier, Alexandre, et al. "Deepsvg: A hierarchical generative network for vector graphics animation." Advances in Neural Information Processing Systems 33 (2020): 16351-16361.
>
> [2] Ganin, Yaroslav, et al. "Computer-aided design as language." Advances in Neural Information Processing Systems 34 (2021): 5885-5897.
>
> [3]Liu, Hanyuan, et al. "End-to-end Line Drawing Vectorization." (2022).
>
> [4]Yamaguchi, Kota. "CanvasVAE: Learning to Generate Vector Graphic Documents." Proceedings of the IEEE/CVF International Conference on Computer Vision. 2021.
>
> # Q2. Can the model extend to multiple mapping sessions?
> >The proposed approach works only with a single-pass mapping of a particular area. How would multiple mapping sessions be integrated to obtain higher-quality maps?
>
> Our paper focuses on modeling the HD map for autonomous driving in real time. Multiple-session fusion is a great idea, so we make an initial attempt at it according to your suggestion.
>
> The nuScenes dataset is composed of multiple traversals collected from different times with geographical overlaps. We aggregate multiple traversals of the same location to enhance the BEV features of VectorMapNet. We use a gated recurrent unit(GRU) to fuse past BEV features with current ones. For each sample, its BEV feature approximately fuses 1-6 past BEV features (the exact number depends on the dataset) from past traversals. The results of the feature aggregation are shown in the table below. We find that VectorMapNet benefits from historical observations.
>
> | method                   | Fuse historical observations | $AP_{Divider}$ | $AP_{Crossing}$ | $AP_{Boundary}$ | Mean AP |
> |--------------------------|------------------------------|------------|-------------|-------------|---------|
> | VectorMapNet Camera only | No                           | 47.3       | 36.1        | 39.3        | 40.9    |
> | VectorMapNet Camera only | Yes                          | **49.6**       | **42.9**        | **41.9**        | **44.8**    |
>
> # Q3. Computation requirement
> >There is no mention of the computational requirements and runtime of the approach. Does it work in real-time? Can it be run on an edge node or does it have to run on more powerful machines in the cloud?
>
> We tested VectorMapNet's inference speed compared to HDMapNet on 1 Intel Xeon Platinum 8358 CPU @ 2.60GHz + 1 RTX 3090 GPU, as shown in the table below.
> VectorMapNet consumes 2.5Gb memory in the inference stage at approximately 8 frames per second for a single model. We find that the computational bottleneck is the polyline generator, and the polyline generator and map element detector can be further optimized to work asynchronously. So we believe that in practical applications, with additional optimization, VectorMapNet is able to run in real-time (typically 10 FPS for self-driving) on an advanced onboard GPU, like Orin.
>
> | Model        | Frame per second (FPS) | Speedup |
> |--------------|------------------------|---------|
> | HDMapNet     | 0.5                    | 1x      |
> | VectorMapNet | 8.3                    | 16.6x   |
>
> We thank Reviewer TD4k for their review of our manuscript again, and we hope that the above responses adequately address all the concerns.

---

### Author Response · Authors · 2022-11-16
**General Response**

Dear Area chair and Reviewers.
Thank you for your hard work and constructive feedback. We appreciate that all the reviewers have acknowledged our work's technical and empirical novelty.  And we were encouraged to hear that the reviewers found our model interesting (reviewer TD4k and oajd), addressing an important problem in self-driving (reviewer sk1h) and novel (reviewer 5FhD).

We have uploaded a new version of the manuscript (revisions are highlighted by blue) incorporating reviewers’ comments. In summary, we: (1) updated the task formulation and polyline definition in Sec. 2 to improve clarity; (2) added a new subsection (Sec. 4.4) to discuss related works about learning vector representations from images; (3). clarified evaluation protocol in Sec. 3.

Furthermore, to ensure our work is reproducible and help you to understand our responses better, we have uploaded our code and some visualization in the supplementary material:
1. VectorMapNet codebase.
2. Visualizations of map prediction with centerlines.
3. The plot of polyline vertex number distribution.
4. An example of expected polyline map outputs.

However, we find that some reviewers are confused with our task setting and model. So our response is organized into the following three parts:
  1. Our problem setup.
  2. Details of the polyline definition.
  3. The evaluation protocol.

## 1. HD Map learning problem setup
The HD map learning task in our paper follows the settings proposed in HDMapNet [1]. Concretely, this task requires the model to detect three types of map elements (road boundaries, lane dividers, and pedestrian crossings) in urban environments with only onboard sensors:

  1. Road boundaries, boundaries of roads that split roads and sidewalks. Typically, they are curves with irregular shapes and arbitrary lengths.
  2. Lane dividers, boundaries of the lanes in the road. Usually they are straight lines.
  3. Pedestrian crossings, regions with white markings where pedestrians can legally cross the road. Usually they are quadrilaterals.

These map elements have various geometrical shapes. Previous works represent them in raster maps, which have multiple drawbacks, as described in the introduction. As mentioned by **reviewer TD4k**, almost all modern commercial mapping software and most autonomous driving datasets, such as the nuScenes and the Argoverse2 dataset, represent maps in vectorized form. These vector maps are usually annotated by human labor. Our goal is to predict these vector maps.

It is important to point out that our work formulate this task as **a detection problem, not a segmentation problem or a contour extraction problem**. Our approach first detects map elements, then refines their shapes, in a top-down manner. Finally, our model predicts a set of map elements. Each map element is represented as a polyline.

## 2. Polylines Definition
A polyline is a sequence of vertices arranged according to their orders, and the number of vertices is variable. We use polyline as primitive because it can form arbitrary shapes and describe arbitrary geometries, which is an ideal property because map elements may have variant shapes and arbitrary length.

We convert vector HD maps to polylines by applying the Douglas-Peucker algorithm[1].  We also have plotted the polyline vertex number distribution of the nuScenes validation set, which is contained in the supplementary material.

## 3. Evaluation Protocol

**Why use average precision (AP):**

Because we formulate our task as a detection problem. And AP is commonly used for evaluating the quality of model, in instance level.

**Why not use IoU:**

Because our task is not a segmentation problem and it is senseless to use IoU as metric on a detection problem. We do not assign labels to pixels. Furthermore, our task emphasizes the concept of instance, which also cannot be evaluated by IoU.

**Details of mAP evaluation:**

Our evaluation protocol mainly follows the one proposed in HDMapNet [2]. We use the average precision (AP) metric that is commonly used in detection tasks to evaluate prediction performance. The only difference is the selection of distance metric used in TP/FP matching. HDMapNet [2] and our paper use Chamfer distance for matching. Additionally, we also propose another distance metric termed Fréchet distance, which better measures the distance between polylines. More details are discussed in section A.2 of our paper.

When comparing our method with other methods, we use Chamfer distance-based AP. When doing ablation studies of our model, we provide both results of Chamfer distance based AP and Frechet distance based AP.

[1] Saalfeld, Alan. "Topologically consistent line simplification with the Douglas-Peucker algorithm." Cartography and Geographic Information Science 26.1 (1999): 7-18.

[2] HDMapNet: An Online HD Map Construction and Evaluation Framework. arXiv:2107.06307

---

### Decision · Program_Chairs · 2023-01-20

**Decision:**

Reject

**Justification For Why Not Higher Score:**

Said in the meta-review

**Justification For Why Not Lower Score:**

-

**Metareview: Summary, Strengths And Weaknesses:**

This paper has received four expert reviews and was a borderline case. The paper was discussed first asynchronously over openreview and finally a meeting was held between the AC and two of the reviewers (TD4k and sk1h), which had initially given borderline ratings. The discussion of the paper again brought several issues to light:

- On the positive side, the general idea of the paper had been appreciated and R+AC agreed that it has potential.
- There is a lack of positioning with respect to the state of the art, and the exact claims of the paper.
- The comparison with the SOTA has been assessed as unconvincing:
  - if the claim is the novelty of vectorized representations for autonomous driving, then the usefulness of this representation needs to be evaluated and compared. While the authors provided experiments on a downstream task (motion prediction), it does not include any comparison to prior work.
  - if the claim is novelty of the specific method for generating vectorized representations, then this claim needs to be backed up through comparisons with existing work for predicting vectorized representations, be it for BEVs or for other tasks.
  - the method has been compared with 1 other method, which has been identified as being extremely close to the submitted work (and the double blind process makes it hard to draw conclusions); plus another method on one of the datasets only.
- The writing of the paper is not clear enough. While this has not been considered as grounds for rejection alone, it has also weighted into the decision.

For all these reasons, a consensus emerged that this paper is not yet ready and needs to be improved by better experimental comparisons and further backing up of its claims.


**Summary Of Ac-Reviewer Meeting:**

Explained in the Meta-review